# Molecularly Confirmed Pseudomyogenic Hemangioendothelioma with Unusual EGFL7::FOSB Fusion in the Head and Neck Region of an Older Patient

**DOI:** 10.3390/diagnostics14030342

**Published:** 2024-02-05

**Authors:** Dong Ren, Jerry Lou, Katherine Wei, Ibe Ifegwu

**Affiliations:** 1Departments of Pathology and Laboratory Medicine, University of California Irvine Medical Center, Orange, CA 92868, USA; jjlou@hs.uci.edu; 2Departments of Radiology, University of California Irvine Medical Center, Orange, CA 92868, USA; kjwei@hs.uci.edu

**Keywords:** pseudomyogenic hemangioendothelioma, EGFL7::FOSB fusion, head and neck region, elder patients

## Abstract

Pseudomyogenic hemangioendothelioma (PMHE), a rare vascular neoplasm, was first described in 1992 asa fibroma-like variant of epithelioid sarcoma, and would be termed as epithelioid sarcoma-like hemangioendothelioma a decade later due to its significant histologic overlap with epithelioid sarcoma and diffuse cytokeratin expression. PHME is currently defined as a distinct, potentially intermediate malignant, rarely metastasizing neoplasm with vascular/endothelial differentiation. It is characterized by young age (typically less than 40 years old), extremity location (approximately ~80%), and t(7:19) SERPINE1::FOSB fusion as the most common molecular alteration. Herein, we report a case of a 59-year-old male presenting with multifocal lesions, including in the right temporalis muscle, right frontoparietal calvarium, right pterygoid muscles, and right mandibular condyle. Histologic examination of the right temporal lesion revealed a multinodular biphasic lesion composed of sheets and fascicles of elongated spindle and epithelioid cells infiltrating into the adjacent skeletal muscle. Admixed abundant neutrophilic infiltration is noted; however, areas of necrosis, increased mitosis, nuclear atypia, or rhabdomyoblast-like cells are absent. Immunohistochemical (IHC) staining showed that the tumor cells were diffusely and strongly positive for FOSB, pan-cytokeratin (AE1/AE3), CD31, and ERG. Molecular testing demonstrated a t(9:19) EGFL7::FOSB fusion mRNA. This constellation of morphological, IHC and molecular findings was consistent with a diagnosis of PMHE. This is the first reported case of multifocal PMHE with EGFL7::FOSB fusion in the head and neck area of a patient aged more than 50 years old. Since the differential diagnoses for PMHE includes high-grade malignancies with aggressive clinical behavior, coupled with the rare reports of PMHE in the head and neck region, awareness of this tumor in the head and neck region will avoid the misdiagnosis and overtreatment of this entity.

## 1. Introduction

Pseudomyogenic hemangioendothelioma (PMHE) is a rare, distinctive vascular neoplasm of intermediate malignant potential, often arising in skin and superficial and/or deep soft tissue [1]. It can also occur in bones, such as the femur, knee, and patella [2,3,4]. Although PMHE was originally recognized as a distinct entity in 2003 and termed epithelioid sarcoma-like hemangioendothelioma, this tumor was formally included in the World Health Organization (WHO) Classification of Soft Tissue and Bone Tumours in 2013. PMHE demonstrates a male predilection and a wide age range (14 to 80 years old) with a mean of 31 years old [5]. It predominantly presents as multiple painful or painless nodules with no characteristic clinical manifestation. Occasionally, bone pain, blurred vision, or joint weakness have been reported depending on the location of the lesion [6]. The extremities are the most common site, accounting for approximately 80% of PMHE cases, with lower extremities predominance (~54%), followed by the trunk, and rarely the head and neck region [5,6]. In the fifth edition of the WHO Classification of Soft Tissue and Bone Tumours, approximately 5% of PHME occur in the head and neck area [7]. In a case series that examined fifty cases of PMHE, only two cases (4%) affected the head and neck region: one case presented as multifocal lesions only on the nose of a 47-year-old male, and the other case presented as a solitary lesion on the forehead in a 34-year-old male [5]. Rarely, PMHE spontaneously regresses, and most solitary lesions are treated by complete surgical excision [5]. However, therapeutic strategies for inoperative multifocal PMHE are still controversial. Chemotherapeutic drugs, such as mTOR inhibitors, sirolimus, everolimus, and rapamycin, or radiotherapy is generally administered to patients with multifocal PMHE; however, due to the rarity of this entity, the efficacy of this treatment has not been well established in controlling the disease [8,9,10,11]. PMHE generally has an indolent clinical course, but high rates of local recurrence have been reported in more than 50% of cases [5]. Cases of PMHE with lymph node and distant metastasis are rarely reported, including a case with inguinal lymph node metastasis [5] and two cases with multiple distant metastasis, including to lungs, bones, skin and soft tissue, and axilla [5,12]. 

EGF Like Domain Multiple 7 (EGFL7), located on chromosome 9 (9q34.3), is a member of the epidermal growth factor (EGF) family mainly residing in the endothelium of vasculature in almost all the human organs, such as the lungs, heart, retina, and brain [13,14], and is responsible for the proliferation, migration, and sprouting of lymphatic and vascular channels [15]. EFGL7 protein consists of 273 amino acids (29.6 kDa) including two EGF-like domains [16], one binding to Notch and the other harboring calcium-binding subdomain [16,17]. EGF has been extensively demonstrated to play a pivotal role in angiogenesis during embryogenesis and organogenesis [15,18], and is mediated by multiple signaling pathways, including MAPK, integrin, and Notch [19]. 

FOSB belongs to a basic Helix–Loop–Helix (bHLH) family of transcription factors, the same family as FOS, and exerts its biologic function by initiating transcriptional activities in a variety of cellular processes, including cellular differentiation, growth, and apoptosis [20,21]. Aberrant expression of FOSB has also been reported to play an important role in angiogenesis. Li et al. reported that an inhibitor of angiogenesis, dibenzoxazepinone BT2, inhibited endothelial cell proliferation and angiogenesis by inactivating pERK-FosB/VCAM-1 axis [22]. In addition, IJzendoorn and colleagues demonstrated that SERPINE1::FOSB fusion promoted the formation of abnormal vessels by introducing the chromosomal translocation of SERPINE1::FOSB into human induced pluripotent stem cells (hiPSCs), which further facilitated the development of PHME in mice [23]. Interestingly, FOSB overexpression has also been associated with the inflammatory infiltrate in angiolymphoid hyperplasia with eosinophilia, also known as epithelioid hemangioma [24,25]. Hakar and colleagues first reported the gene fusion between EGFL7 RNA transcript on exon 7 and FOSB on exon 2 [12]. Notably, the patient with EGFL7::FOSB fusion was a 12 year old male who presented with widely metastatic disease, including multiple subcutaneous nodules on the scalp and skin lesions on the hands, as well as lung, liver, and brain lesions, suggesting that EGFL7::FOSB fusion may be associated with the aggressive or metastatic behavior of PMHE. However, the biologic function of EGFL7::FOSB fusion in tumorigenesis is still unknown. 

To the best of our knowledge, the case presented below is the first reported case of multifocal PMHE in a patient aged more than 50 years old with EGFL7::FOSB fusion in the head and neck area with widespread distribution, involving the right calvarium, right temporalis muscle, and right mandible.

## 2. Case Presentation

A 59-year-old male presented to the clinic at our institution with a right temporal mass. About six (6) months ago, he began having mild to moderate pain in the right side of his jaw when he opened his mouth or yawned, and subsequently noticed a painless lump about two (2) months later. Past medical history was notable for hypertension, type II diabetes mellitus, hypercholesterolemia, and arthritis. Past surgery history included endoscopic bilateral knee surgery and abdominal hernia surgery. The patient denied smoking or alcohol use, and laboratory values were unremarkable. 

MRI imaging showed a heterogeneous lesion measuring 2.1 × 1.8 × 1.2 cm centered in the right temporalis muscle (Figure 1A). Furthermore, four additional discontiguous smaller lesions within the right frontoparietal calvarium (9 × 8 × 6 mm); the deep aspect of the right temporalis muscle (12 × 9 × 7 mm); and the superficial aspect of the right masseter muscle (11 × 7 × 7 mm); and the right mandibular condyle (9 × 8 × 8 mm) were reported (Figure 1B). All four lesions were greater than three (3) cm away from the largest lesion (Figure 1C). PET CT showed an increased uptake in these lesions concerning neoplasm (Figure 1D). The patient eventually underwent an excisional biopsy of the right temporal mass.

Histologically, the right temporal mass exhibited a multinodular architecture and showed subtle infiltration into the adjacent skeletal muscle. The tumor was composed of sheets and fascicles of elongated spindle cells and round to ovoid epithelioid cells. Within the epithelioid component, abundant neutrophils are noted (Figure 2). The tumor cells displayed vesicular nuclei and variable nucleoli, and areas of microhemorrhages were scattered throughout the lesion. Nuclear pleomorphism, mitotic figures, necrosis, and rhabdomyoblast-like cells were not identified. IHC work-up revealed that the tumor cells were diffusely and strongly positive for pan-cytokeratin (AE1/AE3), CD31, and ERG, and negative for CD34 (Figure 3). Other negative stains included HHV8, EMA, TLE1, p40, p63, S100, SOX10, Desmin, and SMA. INI-1 was retained, and Ki-67/Mib1 was highlighted in 15–20% of tumor cells. While the IHC findings were consistent with a vascular neoplasm, it excluded histologic mimics, including epithelioid sarcoma and epithelioid hemangioendothelioma. A FOSB immunohistochemical stain was performed, and the tumor cells were diffusely positive (Figure 3), which was supportive of a diagnosis of PMHE. Since FOSB positivity has been reported in other vascular neoplasms such as epithelioid hemangioma [26,27], DNA and RNA next-generation sequencing (NGS) was performed. The NGS results showed a t(9:19) EGFL7::FOSB fusion mRNA, consistent with the genomic alteration of PHME. Based on the constellation of histological, IHC, and molecular findings, the pathologic diagnosis was consistent with pseudomyogenic hemangioendothelioma (PMHE) with EGFL7::FOSB fusion. 

Six (6) months later, the patient underwent an excisional biopsy of the right frontoparietal calvarium lesion. This lesion showed similar morphological findings to the patient’s first excisional biopsy (the right temporal mass), and was consistent with a diagnosis of metastatic PMHE (Figure 4). The patient had an uneventful postoperative course and is currently receiving radiation therapy for his other lesions. 

## 3. Discussion

Vascular lesions are found along the spectrum from benign (hemangioma) to malignant (angiosarcoma). Although benign and malignant vascular tumors are more common, some rare vascular tumors show a biologic behavior that is intermediate between hemangiomas and angiosarcomas, and are termed hemangioendotheliomas. Examples include epithelioid hemangioendothelioma (EHE), pseudomyogenic hemangioendothelioma (PMHE), kaposiform hemangioendothelioma, amongst others [1].

Histologically, PMHE is predominantly composed of sheets and fascicles of elongated spindle cells with abundant eosinophilic cytoplasm, vesicular nuclei, and variable nucleoli. Mild to moderate cytologic atypia can be present, but it is rarely marked. Epithelioid cells may be present in PMHE and can be intimately intermingled with the spindle cell components. In a series of 50 PMHE cases, focal epithelioid cells were identified in all cases [5,6]. Rarely, prominent biphasic spindle and epithelioid components are clearly demarcated from each other. Rhabdomyoblast-like cells and neutrophils infiltration are variably present. Mitoses are scarce. Foci of necrosis have been reported in 14.5% of PMHE cases in one case series (8/55) [5,28]. The lesion can show an infiltrative margin, and vascular differentiation/vasoformation may not be evident until immunohistochemical stains are applied.

Due to its non-specific morphological features, a broad spectrum of differential diagnoses, including benign and malignant tumors, is typically considered before the diagnosis of PMHE. Benign entities such as nodular fasciitis and fibrous histiocytoma, and malignant tumors including epithelioid sarcoma, leiomyosarcoma, spindle cell squamous cell carcinoma, spindle cell melanoma, monophasic spindle cell synovial sarcoma, spindle cell rhabdomyosarcoma, angiosarcoma, and epithelioid hemangioendothelioma (EHE), are histologic mimics. Therefore, in conjunction with the histomorphology, a comprehensive immunohistochemical and molecular testing panel is needed to exclude the differential diagnoses.

Immunohistochemically, all PMHE cases have been reported to show positivity for pan-cytokeratin (AE1/AE3), FLI-1, ERG and INI-1, and negativity for CD34. The tumor is positive for CD31 (40–50%) and demonstrates variable reactivity to CAM5.2, EMA, and SMA [5,6]. The most distinct form of molecular signature in PMHE is a genetic fusion between FOSB (19q) and SERPINE (7q22) [29,30]. Furthermore, WWTR1(3q25), a common partner fusion gene with CAMTA1 in epithelioid hemangioendothelioma, [31,32] has also been reported to be a partner fusion gene of FOSB in PMHE [33]. Other gene fusions, such as t(7:19) ACTB::FOSB and t(2:19) POTEI::FOSB, have been reported in PMHE as well [34,35]. All these fusions upregulate FOSB expression, which is positive in several diseases, including PMHE [36], epithelioid hemangioma [25] and osteoblastomas [37]. In the current study, molecular testing revealed, for the first time, a rare t(9:19) EGFL7::FOSB fusion mRNA in PMHE in the head and neck region. Importantly, although metastasis is extremely uncommon in PMHE [5,38], this novel gene fusion of EGFL7::FOSB was first identified in PMHE from multifocal hand lesions in a widely metastatic disease, including numerous masses in the lungs, liver, soft tissues, and axial and appendicular skeleton, at the initial presentation [12], suggesting that EGFL7::FOSB fusion may be associated with or drive the metastatic potential of PMHE. Interestingly, the patient’s MRI showed an additional four lesions which shared similar radiologic features to the lesion at right temporalis muscle, and an excisional biopsy of the right calvarium lesion confirmed widespread PMHE. 

As mentioned above, PMHE is characterized by young age (94% of patients are less than 50 years old) and extremity location (approximately 80%) [5,6,39]. The head and neck region is one of the rarest sites, and no more than 15 cases, to the best of our knowledge, have been reported in this area so far (Table 1) [5,36,39,40,41,42,43]. Among them, the age of all the patients was less than 50 years old, with the eldest patient being aged 47 years old. In this report, we present a 59-year-old male with multifocal lesions in the head and neck region, and histopathologic and molecular confirmation of a diagnosis of PHME. So far, this is the first reported case of PMHE with EGFL7::FOSB fusion in the head and neck region, as well as the first case in a patient aged more than 50 years old in the head and neck region. 

## 4. Conclusions

In conclusion, to the best of our knowledge, this is the first case to report PMHE with EGFL7::FOSB fusion in the head and neck area. Awareness of this entity will allow clinicians and pathologists to carefully consider PMHE in older patients presenting with lesions exhibiting spindled and epithelioid histomorphology, even in the absence of overt vasoformation. In turn, this will prevent the misdiagnosis and overtreatment of a relatively low-grade neoplasm.

## Figures and Tables

**Figure 1 diagnostics-14-00342-f001:**
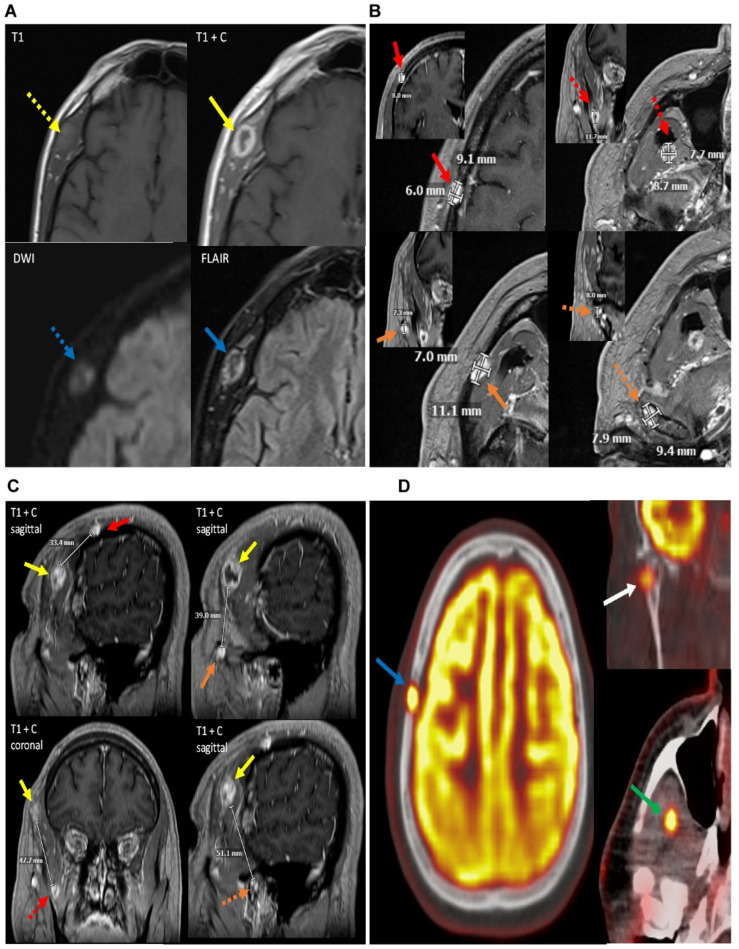
(**A**) Axial images of the patient’s MRI with and without contrast demonstrate a heterogeneous, peripherally enhancing, 2.1 × 1.8 × 1.2 cm (AP × ML × CC) lesion centered within the right temporalis muscle (solid yellow arrow). The lesion demonstrates T1 isointensity to surrounding muscle tissue (dotted yellow arrow). On diffusion weighted sequences, the lesion demonstrates heterogeneous hyperintensity on diffusion restriction sequence (dotted blue arrow), suggesting areas of high cellularity in the lesion. FLAIR (fluid-attenuated inversion recovery) sequences also demonstrate heterogeneously increased signals from the lesion (solid blue arrow). Findings are suggestive of a neoplastic process, with possible cystic change or central necrosis due to the lack of central enhancement. (**B**) Multiple axial and coronal T1 post-contrast sequences on the patient’s brain MRI demonstrate additional lesions with similar imaging features to the right temporalis muscle lesion in Figure 1, including T2/FLAIR (fluid-attenuated inversion recovery) hyperintensity, diffusion restriction, and enhancement. Of note, some of these lesions demonstrate slight differences in enhancement pattern compared to the primary right temporalis muscle lesion, with overall more homogeneous enhancement. This is likely related to differences in size. The lesion in the deep aspect of the temporalis muscle, however, demonstrates an enhancement pattern more similar to the primary lesion, with more heterogeneous peripheral enhancement. These lesions are centered in the right frontoparietal calvarium (red solid arrow) measuring 9 × 8 × 6 mm (AP × ML × CC), deep aspect of the right temporalis muscle (dotted red arrow) measuring 12 × 9 × 7 mm (AP × ML × CC), superficial aspect of the right masseter muscle with possible extension into the right zygomatic arch (solid orange arrow) measuring 11 × 7 × 7 mm (AP × ML × CC), and the right mandibular condyle (dotted orange arrow) measuring 9 × 8 × 8 (AP × ML × CC). (**C**) Contrast-enhanced T1 sequences in the sagittal and coronal planes demonstrate the distance of the primary lesion in the superficial aspect of the right temporalis muscle (solid yellow arrow) from the additional enhancing lesions within the head and neck region. The primary lesion is approximately 3.3 cm from the lesion within the right frontoparietal calvarium (solid red arrow), approximately 3.9 cm from the lesion within the right masseter muscle (solid orange arrow), approximately 4.8 cm from the deep aspect of the right temporalis muscle (dotted red arrow), and approximately 5.1 cm from the right mandibular condyle (dotted orange arrow). (**D**) PET-CT (positron emission tomography computed tomography) scan in the same patient demonstrates increased FDG (fluorodeoxyglucose 18F) radiotracer uptake with maximal uptake of 9.3 SUV in the right frontoparietal calvarial lesion which corresponds to a lytic lesion on the accompanying CT (blue arrow), concerning neoplasm. There are additional FDG avid lesions in the right mandibular condyle with maximal uptake of 5.1 SUV (white arrow) and in the deep aspect of the right temporalis with maximal uptake of 8.7 SUV (green arrow), also concerning for neoplastic process. Previously noted right temporal lesion is not seen on this PET-CT as it was excised, although was shown to be FDG avid on prior PET. The masseter/zygomatic arch lesion demonstrated FDG uptake slightly above background activity at approximately 2.7 SUV and is indeterminate for neoplastic or reactive process.

**Figure 2 diagnostics-14-00342-f002:**
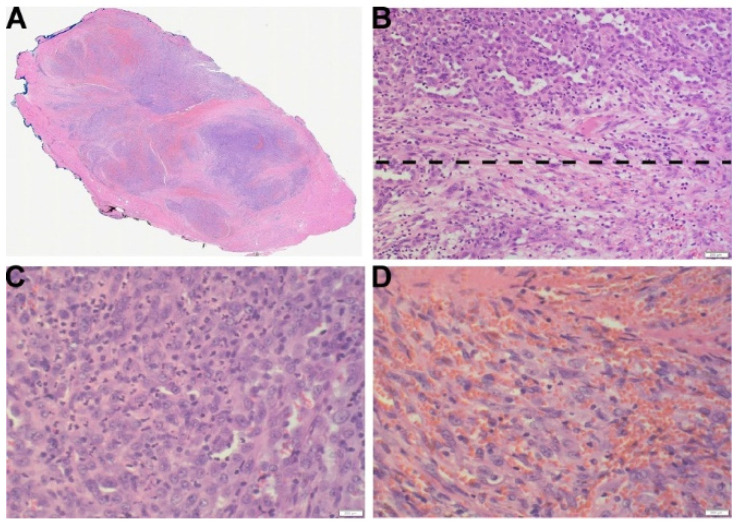
Histologic findings of PMHE lesion at the right temporalis muscle. (**A**) Section of the final resection specimen from right temporal mass shows a multinodular lesion separated by fibrous septa, 4×. (**B**) A well-demarcated biphasic lesion is composed of sheets and fascicles of elongated spindle cells (spindle cell component-lower part of the dotted line), and epithelioid small round to ovoid cells with abundant eosinophilic cytoplasm (epithelioid component-upper part of the dotted line), 20×. (**C**) Representative section of epithelioid component with neutrophilic infiltration, 40×. (**D**) Representative section of spindle cell component with neutrophilic infiltration, 40×.

**Figure 3 diagnostics-14-00342-f003:**
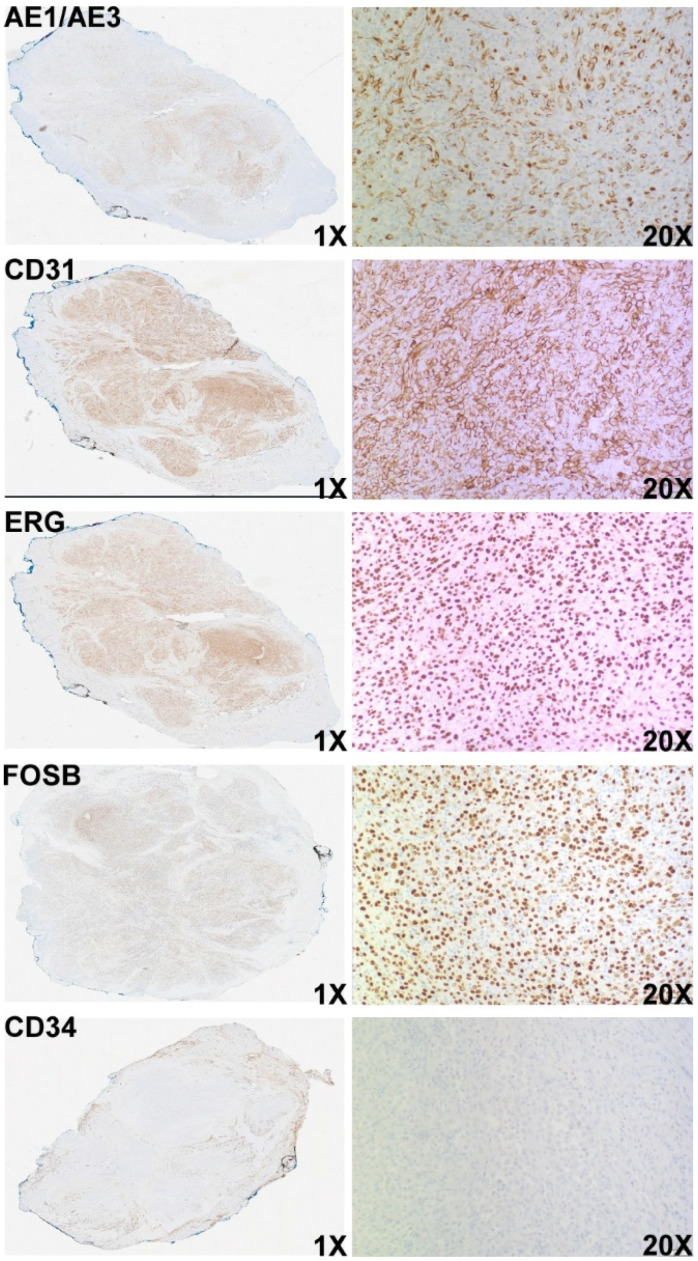
Immunostains of the lesion reveal that the tumor cells are diffusely positive forAE1/AE3, CD31, ERG, and FOSB, but negative for CD34.

**Figure 4 diagnostics-14-00342-f004:**
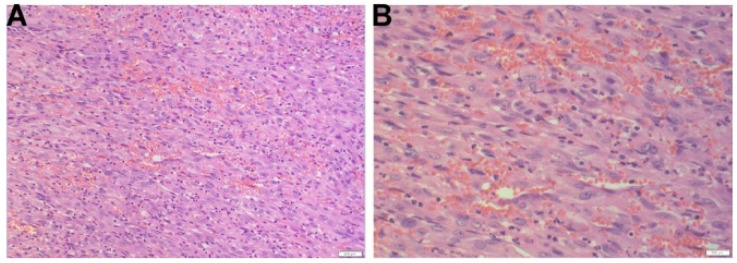
Histologic findings of PMHE at the right frontoparietal calvarium. (**A**,**B**) The lesion is predominantly composed of epithelioid small round to ovoid cells with abundant eosinophilic cytoplasm admixed with sheets and fascicles of elongated spindle cells, 20× (**A**) and 40× (**B**).

**Table 1 diagnostics-14-00342-t001:** Summary list of reported PMHE in the existing literature.

Frequency	Site	Literature
2	Skull	Hung et al. (2017) [36]
2	Nose	Hornick et al. (2011) [5], Mittal (2022) [43]
2	Face	Hung et al. (2017) [36]
2	Oral cavity	Rawal et al. (2017) [39], Requena et al. (2013) [41]
1	Scalp	Mittal (2022) [43]
1	Forehead	Hornick et al. (2011) [5]
1	Upper lip	Requena et al. (2013) [41]
1	Neck	Cai et al. (2011) [42]
1	Multiple skin on the head and neck	IJzendoorn et al. (2018) [40]
1	Intra-parotid lymph node	Hung et al. (2017) [36]

## Data Availability

Data are contained within the article.

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
