# Peer review of "Molecularly Confirmed Pseudomyogenic Hemangioendothelioma with Unusual EGFL7::FOSB Fusion in the Head and Neck Region of an Older Patient"

_diagnostics, 2024, doi:10.3390/diagnostics14030342_

Round 1

Reviewer 1 Report

Comments and Suggestions for Authors

There are several areas in the manuscript that need to be improved:

1. Introduction:

i) More information on the biologic behavior of this tumor is needed.

ii) Also include information on the diagnostic histologic and molecular features.  

iii) What is the role of EGFL7/FOSB fusion in tumor biology and in driving tumor formation

2. Case presentation:

i) Please include 3 measurement dimensions for the MRI results. If the lesion is spherical, you can mention one dimension as a diameter.  

ii) Measurements should correspond to the measurements in figure 1 legend, where 1.8 x 1.2 x 2.1 cm is cited versus 1.6 x 1.1 cm in the text

iii) What was the histologic differential diagnosis and the reason NGS testing was specifically performed for EGFL7/FOSB fusion.

iv) Please include information on how the patient was treated and any follow-up information

3. Discussion:

i) Discussion is poor and includes repetition of case results and other information that should be included in the case report (lines 140-155).   

ii) Line 173 suggests that the other lesions could be metastasis of the primary temporalis lesion. To illustrate this suggestion, the authors should present an MRI image or illustration showing the exact positions of all lesions in the head and neck area and how far they are located from one another.

iii) Authors should include a table summarizing all the relevant PNHE cases reported in the head and neck, to support their discussion in lines 178-181

4. Figures

i) Histologic figures are low magnification and not clear. PLease include high power images of the histology and positive immunostains

 5) Methods

i) Methods section is needed that summarizes how the tissue is handled

ii) Details on the immunostains are needed including instrumentation. Are immunostains performed in a clinical lab? FOSB immunostain is relatively unknown.  Please include further details on the primary antibody, source and positive control

iii) Details on the molecular tests performed should be included: the nature of the NGS method, whole exome or targeted panel, the RNA fusion test methodology, etc. . Was the NGS performed in a CLIA-approved laboratory? Any additional information on other molecular findings?

Comments on the Quality of English Language

English language needs to be revised and adjusted in several areas:

Title: Molecularly confirmed ---

line 40; are

line 41: accounting, not occupying for

line 52: "jay", of yawned?

line 144: diffusely positive

line 179: till date?

Reviewer 2 Report

Comments and Suggestions for Authors

The clinic case that the authors present is interesting and very rare in the head and neck region. The molecular and IHC studies support the diagnosis made, and the authors describe this case well. They conducted essential molecular studies to arrive at the diagnosis, and the case was well described. Despite the interesting study and case presented, I have some suggestions that the authors need to review.

1. Would it be possible to review the WHO classification of soft tissue and bone tumors (5th ed.) 2020?

2. Interestingly, this soft tissue tumor is not classified in Section Soft Tissues of the Head and Neck Tumors (5th ed.), 2022. Would it be possible to give some hypotheses about that?

3. According to the WHO classification, the PMHE of the head and neck is approximately 5%. Would it be possible to review that and include the WHO classification of soft tissue and bone tumors (5th ed.) 2020 in their references?

4. This is a rare case, mainly in the head and neck region, and the EGFL7::FOSB fusion is most rare. What is your opinion, according to this case and the molecular alteration reported by the WHO that only describes FOSB?.

Round 2

Reviewer 1 Report

Comments and Suggestions for Authors

I do not have any further comments